# Diagnostic Accuracy of ki-67 Labeling Index in Endoscopic Ultrasonography-Fine-Needle Aspiration Cytology and Biopsy of Pancreatic Neuroendocrine Neoplasms

**DOI:** 10.3390/diagnostics13172756

**Published:** 2023-08-25

**Authors:** Jung-Soo Pyo, Nae Yu Kim, Kyueng-Whan Min, Il Hwan Oh, Dae Hyun Lim, Byoung Kwan Son

**Affiliations:** 1Department of Pathology, Uijeongbu Eulji Medical Center, Eulji University School of Medicine, Uijeongbu-si 11759, Gyeonggi-do, Republic of Korea; jspyo@eulji.ac.kr (J.-S.P.); kyueng@eulji.ac.kr (K.-W.M.); 2Department of Internal Medicine, Uijeongbu Eulji Medical Center, Eulji University School of Medicine, Uijeongbu-si 11759, Gyeonggi-do, Republic of Korea; naeyu46@eulji.ac.kr (N.Y.K.); 20180121@eulji.ac.kr (I.H.O.); daehyun.lim@eulji.ac.kr (D.H.L.)

**Keywords:** pancreatic neuroendocrine neoplasm, endoscopic ultrasonography, fine needle aspiration cytology/biopsy, Ki-67, meta-analysis, diagnostic test accuracy review

## Abstract

Background: This study aimed to compare the diagnostic accuracy of the Ki-67 labeling index (LI) between endoscopic ultrasonography-fine-needle aspiration cytology/biopsy (EUS-FNAC/FNB) and surgical specimens of pancreatic neuroendocrine neoplasms (PanNENs). Methods: Conventional meta-analysis and diagnostic test accuracy (DTA) reviews were performed on 17 eligible studies. The DTA review involved calculating the sensitivity, specificity, diagnostic odds ratio (OR), and area under the curve (AUC) of the summary receiver operating characteristic (SROC) curve. In addition, subgroup analysis was conducted based on EUS-FNAC and FNB, tumor grade, and tumor size. Results: The overall concordance rate of WHO grade based on Ki-67 LI between the EUS-FNAC/FNB and the surgical specimen was 0.767 (95% confidence interval (CI), 0.713–0.814). Concordance rates of the EUS-FNAC and EUS-FNB subgroups were 0.741 (95% CI, 0.681–0.794) and 0.839 (95% CI, 0.738–0.906), respectively. In the DTA review for grade 3, the sensitivity and specificity were calculated to be 0.786 (95% CI, 0.590–0.917) and 0.998 (95% CI, 0.987–1.000), respectively. The diagnostic OR and AUC of the SROC curve were 150.220 (95% CI, 46.145–489.000) and 0.983, respectively. The sensitivity and specificity were observed to be highest in the grade 1 and 3 subgroups, respectively. Conclusions: Higher concordance of tumor grade based on Ki-67 LI was observed between EUS-FNAC/FNB and surgical specimens, indicating the potential usefulness of Ki-67 LI in predicting PanNEN tumor grade in EUS-FNAC/FNB.

## 1. Introduction

Pancreatic neuroendocrine neoplasms (PanNENs) account for 2–5% of all pancreatic neoplasms [1,2]. They are classified into well-differentiated and poorly differentiated NENs [1]. The grade of PanNENs is determined based on the mitotic rate and Ki-67 labeling index (LI). While most PanNENs exhibit indolent behavior, there are instances of recurrence [3,4]. The recurrence of PanNENs is associated with factors, such as tumor size, Ki-67 LI, and tumor grade [4,5]. Endoscopic ultrasonography (EUS) is a commonly used preoperative diagnostic tool for evaluating pancreatic lesions. PanNEN specimens were usually collected using an EUS-fine-needle aspiration cytology/fine-needle biopsy (EUS-FNAC/FNB) [6]. The tumor grade of PanNENs can be assessed using the EUS-FNAC/FNB specimens. Despite a relatively strong correlation between the preoperative tumor grade and the tumor grade of the surgical specimens [7,8,9,10,11,12,13,14,15,16,17,18,19,20,21,22,23], the assessment of tumor grade based on EUS-FNAC/FNB samples is challenging in daily practice due to limitations in sample volume or cellularity. The diagnosis and grading of PanNENs can be particularly challenging when dealing with smaller tumor sizes [24,25]. Accurate diagnosis and tumor grading through EUS-FNAC/FNB can significantly influence the choice of treatment modality. The assessment of Ki-67 LI assessment using EUS-FNAC/FNB plays a vital role in the accuracy and usefulness of preoperative examinations. The accuracy of Ki-67 LI diagnosis has been investigated in several studies, but the sample size of PanNENs in each study was limited [7,8,9,10,11,12,13,14,15,16,17,18,19,20,21,22,23]. Furthermore, the concordance between Ki-67 LI in EUS-FNAC/FNB and surgical specimens has shown variability across different reports [7,8,9,10,11,12,13,14,15,16,17,18,19,20,21,22,23]. Given the limitations of individual studies, a meta-analysis can offer a more comprehensive assessment of the diagnostic accuracy of Ki-67 LI in PanNENs, providing a robust evaluation of its diagnostic value.

The objective of this study was to assess the diagnostic test accuracy of the Ki-67 LI in EUS-FNAC/FNB for PanNENs using a meta-analysis and diagnostic test accuracy (DTA) review approach. The study examined the concordance rates of Ki-67 LI between EUS-FNAC/FNB and surgical specimens and performed a subgroup analysis based on tumor grade, sampling method, and tumor size.

## 2. Materials and Methods

### 2.1. Published Study Search and Selection Criteria

Relevant articles were obtained by searching the PubMed databases through 20 January 2023. Searching was performed using the following keywords: “(pancreas or pancreatic) AND (EUS or endoscopic ultrasound or aspiration cytology) AND (Ki-67 or Ki67 or proliferation index)”. The titles and abstracts of all searched articles were screened for exclusion. Review articles and previous meta-analyses were also screened to obtain additional eligible studies. Searched results were then reviewed, and articles were included if the study investigated the PanNENs and there was information for Ki-67 immunohistochemistry. The articles that were case reports or non-original articles, or non-English language publications were excluded.

### 2.2. Data Extraction

Data from all eligible studies were extracted by two individual authors. Extracted data from each of the eligible studies included the following [7,8,9,10,11,12,13,14,15,16,17,18,19,20,21,22,23]: first author’s name, year of publication, study location, number of patients analyzed, types of obtained EUS samples, tumor grade of aspiration and surgical specimen, and so on. For the meta-analysis and diagnostic test accuracy review, all data associated with the Ki-67 LI of PanNEN were extracted. Any disagreements were resolved by consensus.

### 2.3. Statistical Analysis

For performing the meta-analysis, all data were analyzed using the Comprehensive Meta-Analysis software package (Biostat, Englewood, NJ, USA). The concordance rates of tumor grade by Ki-67 labeling index were investigated between EUS-FNAC/FNB and surgical specimens. In addition, the subgroup analysis based on types of obtained EUS samples, tumor grade, and tumor size was performed. Because the eligible studies used various evaluation methods for Ki-67 IHC of PanNEN and had different populations of patients, a random-effects model was more appropriate than a fixed-effects model. Heterogeneity between the eligible studies was checked using P statistics (*p*-value). In addition, the significance of difference between subgroups was evaluated using a meta-regression test. To evaluate publication bias, Begg’s funnel plot and Egger’s test were conducted. The results with *p* < 0.05 were considered statistically significant. If significant publication bias was found, the fail-safe N and trim-fill tests were additionally conducted to confirm the degree of publication bias. The results were considered statistically significant with *p* < 0.05.

In the present study, the diagnostic test accuracy (DTA) review of the Ki-67 labeling index of PanNEN based on tumor grade was performed using the Meta-Disc program (version 1.4; Biostatics, the Ramon y Cajal Hospital, Madrid, Spain) [26]. The pooled sensitivity and specificity, diagnostic odds ratio (OR) were calculated from individual data of each eligible study. By plotting ‘sensitivity’ and ‘1-specificity’ of each study, the SROC curve (summary receiver operating characteristic curve) was firstly constructed and the curve fitting was performed through linear regression using the Littenberg and Moses linear model [27]. Because each of the data were heterogeneous, the accuracy data were pooled by fitting a SROC curve and measuring the value of the area under the curve (AUC) [26]. An area under the curve (AUC) close to 1 means the test is strong, and close to 0.5 means the test is considered poor. Subgroup analysis based on the PanNEN grade was conducted.

## 3. Results

### 3.1. Subsection

#### Selection and Characteristics

Through database searching, 134 articles were identified. We excluded 76 articles of the searched studies through the screening. These 76 articles were excluded due to non-original articles (n = 42), no inclusion or insufficient information (n = 17), studies for other diseases (n = 14), non-English articles (n = 2), and non-human study (n = 1). Next, we reviewed the full text of 58 articles. Forty-one articles were excluded due to no inclusion or insufficient information. Finally, 17 articles were included in this meta-analysis and DTA review (Figure 1 and Table 1).

### 3.2. Concordance Rate of Tumor Grade of Pancreatic Neuroendocrine Neoplasm between EUS-FNAC/FNB and Surgical Specimen

First, the overall concordance rate was investigated between EUS-FNAC/FNB and surgical specimens. The estimated concordance rate was 0.767 (95% CI 0.713–0.814) (Table 2). In a subgroup analysis based on EUS sampling, the concordance rates of FNAC and FNB were 0.741 (95% CI 0.681–0.794) and 0.839 (95% CI 0.738–0.906), respectively. Although the concordance rate of EUS-FNB was higher than that of EUS-FNAC, there was no statistical significance (*p* = 0.071 in the meta-regression test). Next, the subgroup analysis based on the tumor grade of PanNEN was performed. The concordance rates of grade 1/2 and grade 3 subgroups were 0.772 (95% CI 0.722–0.816) and 0.743 (95% CI 0.628–0.945), respectively. In the grade 1/2 subgroup, the concordance rates of EUS-FNAC and EUS-FNB were 0.745 (95% CI 0.695–0.789) and 0.846 (95% CI 0.722–0.921), respectively. The concordance rates of grade 1 and 2 PanNENs were 0.772 (95% CI 0.712–0.820) and 0.741 (95% CI 0.655–0.812), respectively. In addition, in the grade 3 subgroup, the concordance rates of EUS-FNAC and EUS-FNB were 0.879 (95% CI 0.660–0.965) and 0.667 (95% CI 0.154–0.957), respectively. The concordance rate of grade 3 was lower in EUS-FNB than in EUS-FNAC. However, the statistical significance between EUS-FNAC and EUS-FNB was not found in the subgroup analysis (*p* = 0.356 in the meta-regression test). In PanNEN with less than 2 cm, the concordance rate was 0.797 (95% CI 0.726–0.853). In the subgroup analysis, the concordance rates of grade 1 and 2 PanNENs were 0.877 (95% CI 0.791–0.930) and 0.685 (95% CI 0.414–0.870), respectively. There was a significant difference of concordance rates between grade 1 and 2 subgroups in PanNENs with less than 2 cm (*p* = 0.021 in the meta-regression test).

### 3.3. Diagnostic Test Accuracy Review of Ki-67 in the Grading of Pancreatic Neuroendocrine Neoplasm

A DTA review was performed for the sensitivity, specificity, diagnostic OR, and AUC on SROC of EUS-FNAC/FNB compared to the surgical specimen. We evaluated the diagnostic accuracy of EUS-FNAC/FNB in predicting the tumor grade using Ki-67. The comparison between grade 1/2 and 3 subgroups was performed. In grade 3 subgroup, the pooled sensitivity and specificity were 0.786 (95% CI 0.590–0.917) and 0.998 (95% CI 0.987–1.000), respectively (Table 3). The diagnostic OR and AUC on SROC in grade 3 subgroup were 150.220 and 0.983, respectively. In the subgroup analysis based on tumor grade, the pooled sensitivities of PanNEN grades 1 and 2 were 0.908 (95% CI 0.876–0.937) and 0.599 (95% CI 0.534–0.661), respectively. The pooled specificities of PanNEN grades 1 and 2 were 0.616 (95% CI 0.557–0.674) and 0.904 (95% CI 0.872–0.930), respectively. The diagnostic ORs were 14.467 (95% CI 8.892–23.536) and 13.971 (95% CI 8.364–23.335) in PanNEN grades 1 and 2 subgroups, respectively. In addition, the AUC on SROC were 0.871 and 0.859 in PanNEN grades 1 and 2 subgroups, respectively. In PanNEN with less than 2 cm, the pooled sensitivities of PanNEN grades 1 and 2 were 0.852 (95% CI 0.771–0.913) and 0.667 (95% CI 0.498–0.809), respectively. The pooled specificities of PanNEN grades 1 and 2 were 0.675 (95% CI 0.509–0.814) and 0.844 (95% CI 0.762–0.906), respectively. In PanNEN grade 1 subgroup, the diagnostic OR and AUC on SROC were 15.319 (95% CI 5.915–39.677) and 0.841, respectively. In PanNEN grade 2 subgroup, the diagnostic OR and AUC on SROC were 13.093 (95% CI 5.143–33.332) and 0.834, respectively.

## 4. Discussion

To the best of our knowledge, this study represents the first meta-analysis and DTA review focusing on the assessment of tumor grading using Ki-67 LI between EUS-FNAC/FNB and surgical specimens of PanNENs. The increasing number of incidentally detected PanNENs can be attributed to advancements in imaging technology and improved opportunities for evaluation [23]. Treatment decisions for non-functioning PanNENs can pose significant challenges. The treatment of PanNENs is determined on the basis of their size and grade. The histological grade of PanNENs is determined by mitosis and Ki-67 LI. According to the guidelines, PanNENs are classified into three categories, grades 1 to 3, based on Ki-67 LI. When PanNEN is suspected during EUS-FNAC/FNB, the measurement of Ki-67 LI can provide valuable information for guiding treatment decisions. Given the importance of Ki-67 LI on EUS-FNAC/FNB in treatment decision-making, it is imperative to determine the concordance between Ki-67 LI results from EUS-FNAC/FNB and surgical specimens. However, the limited information available from individual studies examining this concordance emphasizes the significance of our results.

The advancements in EUS for diagnosing pancreatic lesions have significantly improved its diagnostic accuracy. However, the evaluation of specimens obtained through EUS-FNAC/FNB remains a challenging aspect. In the case of PanNENs, EUS-FNAC/FNB is commonly regarded as the most accurate and safe diagnostic method. Ki-67 LI is essential for tumor grading in PanNENs, specifically in EUS-FNAC/FNB specimens. The WHO classifies PanNENs into grades 1 to 3, with each grade determined based on mitotic rate and Ki-67 LI. The concordance between EUS-FNAC/FNB and surgical specimens can be utilized to evaluate the diagnostic accuracy of PanNEN grading. Notably, the reported concordance rates exhibit variations across different studies [7,8,9,10,11,12,13,14,15,16,17,18,19,20,21,22,23]. The concordance rates of individual studies ranged from 50.0 to 100.0% in our meta-analysis [7,8,9,10,11,12,13,14,15,16,17,18,19,20,21,22,23]. The tumor grade of PanNENs, as determined from surgical specimens, is considered definitive. However, in situations where surgical removal is not feasible, preoperative EUS-FNAC/FNB specimens can be used to assess Ki-67 LI, providing valuable information for tumor grading. The advancements in EUS techniques have increased the importance of assessing Ki-67 LI in preoperative specimens. However, it should be noted that the assessment of concordance in Ki-67 LI between preoperative and operative specimens may be limited to operable patients only. Therefore, obtaining comprehensive information from individual studies may be challenging. Performing a meta-analysis along with a DTA review can provide a comprehensive assessment of diagnostic accuracy. We evaluated the accuracy of Ki-67 LI in assessing the tumor grade of PanNEN using EUS-FNAC/FNB through a DTA review. We specifically explored the potential benefits of using a core biopsy needle (FNB) for sampling, which may help overcome the limitations associated with the FNA technique. A previous meta-analysis indicated the superiority of FNB over FNA in evaluating pancreatic masses [28]. However, there is a lack of comprehensive data on the concordance of Ki-67 LI between FNAC and FNB in previous studies. Although not statistically significant, FNB exhibited a slightly higher concordance rate compared to FNAC (0.839 vs. 0.741; *p* = 0.071 in the meta-regression test). Interestingly, in the grade 3 subgroup, FNAC showed a higher concordance rate compared to FNB (0.879 vs. 0.667). Notably, the concordance rate of EUS-FNAC in the grade 3 subgroup was significantly higher compared to that in grade 1 and 2, or EUS-FNB subgroups. However, in routine clinical practice, the difference in concordance rates may have limited significance as FNAC and FNB are typically performed simultaneously.

If the tumor size is less than 2 cm, it is classified as low-stage, which has implications for treatment decision-making. Guidelines recommend a wait-and-see approach for asymptomatic patients with tumors less than 2 cm [5,29]. However, there may be a mismatch between guideline recommendations and their application in clinical practice [15]. Previous studies have examined the concordance of Ki-67 LI in PanNENs less than 2 cm. In this meta-analysis, we further analyzed PanNENs less than 2 cm in size. The concordance rate for tumors less than 2 cm was 0.797 (95% CI, 0.726–0.853). Subgroup analysis based on tumor grade yielded concordance rates of grades 1 and 2 of 0.877 (95% CI 0.791–0.930) and 0.685 (95% CI 0.414–0.870), respectively. There was a significant difference between the grade 1 and 2 subgroups (*p* = 0.021 in the meta-regression test). In clinical practice, this finding can be useful for evaluating tumor grading in PanNENs smaller than 2 cm. The DTA review showed higher sensitivity and lower specificity in the grade 1 subgroup compared to the grade 2 subgroup. However, the diagnostic OR and AUC of the SROC were not significantly different between grades 1 and 2.

Discrepancies between EUS-FNAC/FNB and surgical specimens can be attributed to various factors. One important factor is tumor heterogeneity. According to a previous report, grade 2 PanNENs exhibit a greater heterogeneity compared to grade 1 and 3 tumors [9]. Hasegawa et al. demonstrated that the dispersion of Ki- 67 LI was significantly higher in grade 2 tumors compared to grade 1 tumors [30]. Considering the limitations of EUS-FNAC/FNB, it may not be possible to examine the entire lesion, and the evaluation of hotspots can be limited due to the limited number of specimens. Ki-67 LI should be evaluated in at least 500 tumor cells in the hotspot area for accurate tumor grading. Thus, the number of specimens obtained with EUS-FNAC/FNB can impact the concordance rates as it may limit the evaluation of a sufficient number of tumor cells. The following guidelines provided by the World Health Organization (WHO) [1] and the European Neuroendocrine Tumor Society [5] outline the criteria for tumor grading based on Ki-67 LI: (1) grade 1, Ki-67 LI < 2%; (2) grade 2, Ki-67 LI between 3 and 20%; and (3) grade 3, Ki-67 > 20%. If the Ki-67 LI value is close to the threshold (2 or 20%), caution should be exercised in interpreting the results. This variability in tumor grading can significantly impact concordance rates.

The current study has several limitations that need to be acknowledged. Firstly, we were unable to evaluate the concordance rate based on EUS-FNAC/FNB cellularity due to insufficient information available in the eligible studies. It is recommended to count in more than 500 cells for accurate assessment. Secondly, a detailed analysis based on the specific clones of Ki-67 antibodies used could not be conducted due to insufficient information. Lastly, the evaluation method of Ki-67 LI, whether it was eyeballing or automated calculations, could not be analyzed in subgroup analysis due to insufficient information.

## 5. Conclusions

In conclusion, our study demonstrates a high level of concordance between Ki-67 LI measured using EUS-FNAC/FNB and surgical specimens in patients with PanNENs. Notably, when considering PanNENs smaller than 2 cm, we observed a significant concordance rate of Ki-67 LI in both the grade 1 and 2 subgroups. The findings of our DTA review, underscore the value of Ki-67 LI assessment through EUS-FNAC/FNB as a reliable method for predicting the tumor grade of PanNENs. However, further study is needed to evaluate the difference in accuracy for low cellularity of EUS-FNAC/FNB.

## Figures and Tables

**Figure 1 diagnostics-13-02756-f001:**
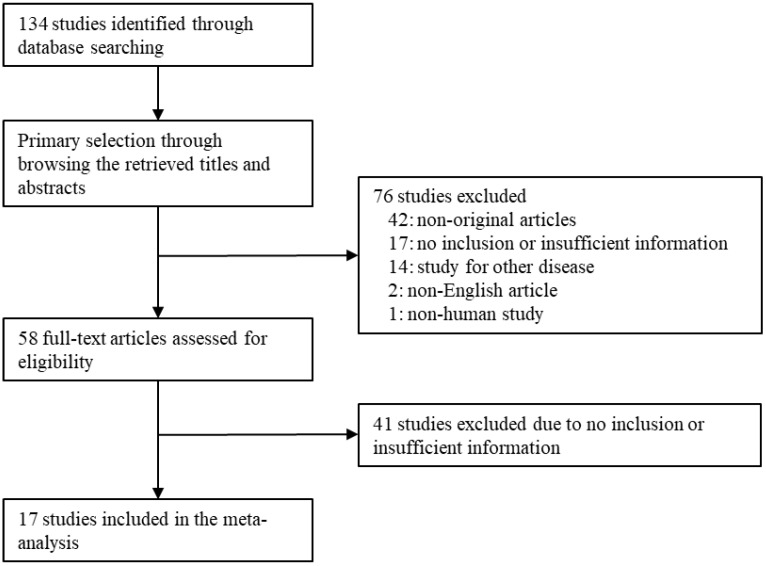
Flow chart of the searching strategy.

**Table 1 diagnostics-13-02756-t001:** Main characteristics of the eligible studies.

Author and Publication Year	Location	Specimen	Antibody Clone	Number of Patients	Surgical Specimen			Tumor Size (cm, Mean ± SD)
					G1	G2	G3	
Abi-Raad 2020 [7]	USA	FNAC with CB	MIB-1	49	27	22	0	3.00 ± 1.90
Boutsen 2018 [8]	Belgium	FNAC	MIB-1	57	30	23	4	2.85 ± 2.16
Crinò 2021 [9]	Italy	FNAC with CB	MIB-1	69	13	4	0	1.98 ± 1.50
		FNB		73	46	25	2	2.15 ± 1.33
Cui 2020 [10]	USA	FNAC	30–9	37	24	8	5	ND
Díaz Del Arco 2016 [11]	Spain	FNAC	ND	10	7	3	0	3.20 ± 3.28
Di Leo 2019 [12]	Italy	FNB	ND	25	20	4	1	2.10 ± 1.49
Farrell 2014 [13]	USA	FNAC	MIB-1	22	15	5	2	3.03 ± 1.73
Grosse 2019 [14]	Austria	FNAC with CB	ND	15	2	9	4	3.84 ± 1.8
Heidsma 2020 [15]	USA	FNAC	ND	63	46	16	1	ND
Hwang 2018 [16]	Korea	FNB		33	20	10	3	3.30 ± 2.20
Kalantri 2020 [17]	India	FNAC with CB	BGX-297	11	4	4	3	ND
Laskiewicz 2018 [18]	USA	FNAC	MIB-1	26	15	11	0	ND
Leeds 2019 [19]	UK	FNAC with CB	ND	23	16	7	0	2.57 ± 0.31
		FNB		26	12	14	0	3.25 ± 0.36
Paiella 2020 [20]	Italy	FNAC	ND	77	48	28	1	2.45 ± 1.34
Piani 2008 [21]	Italy	FNAC	MIB-1	18	11	6	1	3.05 ± 2.77
Sugimoto 2015 [22]	Japan	FNAC	MIB-1	8	5	3	0	2.57 ± 1.32
Tacelli 2021 [23]	Italy	FNAC with CB	MIB-1	112	59	50	3	2.39 ± 0.31

SD, standard deviation; FNAC, fine-needle aspiration cytology; CB, cell block; FNB, fine-needle biopsy; ND, no description.

**Table 2 diagnostics-13-02756-t002:** The estimated concordance rates of WHO grade using ki-67 labeling index between aspirated and surgical specimens in pancreatic neuroendocrine neoplasms.

	Number of Subsets	Fixed Effect [95% CI]	Heterogeneity Test [*p*-Value]	Random Effect [95% CI]	Egger’s Test [*p*-Value]
Overall	19	0.754 [0.719, 0.786]	0.006	0.767 [0.713, 0.814]	0.080
FNAC	15	0.734 [0.694, 0.771]	0.039	0.741 [0.681, 0.794]	0.136
FNB ^a^	4	0.840 [0.770, 0.892]	0.140	0.839 [0.738, 0.906]	0.826
Grade 1/2	19	0.757 [0.722, 0.790]	0.028	0.772 [0.722, 0.816]	0.024
FNAC	15	0.739 [0.699, 0.776]	0.194	0.745 [0.695, 0.789]	0.052
FNB ^b^	4	0.840 [0.766, 0.894]	0.062	0.846 [0.722, 0.921]	0.352
Grade 1	19	0.756 [0.713, 0.794]	0.064	0.772 [0.712, 0.820]	0.026
Grade 2	17	0.732 [0.657, 0.796]	0.310	0.741 [0.655, 0.812]	0.062
Grade 3	6	0.743 [0.628, 0.945]	0.966	0.743 [0.628, 0.945]	0.019
FNAC	5	0.879 [0.660, 0.965]	0.999	0.879 [0.660, 0.965]	<0.001
FNB ^c^	1	0.667 [0.154, 0.957]	1.000	0.667 [0.154, 0.957]	-
Tumor size, less than 2 cm	6	0.797 [0.726, 0.853]	0.777	0.797 [0.726, 0.853]	0.204
Grade 1 ^d^	5	0.877 [0.791, 0.930]	0.939	0.877 [0.791, 0.930]	0.385
Grade 2	3	0.665 [0.453, 0.827]	0.276	0.685 [0.414, 0.870]	0.757

CI, Confidence interval; FNAC, fine-needle aspiration cytology; FNB, fine-needle biopsy. ^a^, *p* = 0.071 in the meta-regression test; ^b^, *p* = 0.063 in the meta-regression test; ^c^, *p* = 0.356 in the meta-regression test; ^d^, *p* = 0.021 in the meta-regression test.

**Table 3 diagnostics-13-02756-t003:** Sensitivity, specificity, diagnostic odds ratio and area under curve of summary receiver operation characteristics curve of evaluating WHO grade using ki-67 labeling index in endoscopic sonography-guided fine-needle aspiration.

	Number of Subsets	Sensitivity (%) [95% CI]	Specificity (%) [95% CI]	Diagnostic OR [95% CI]	AUC on SROC
Overall					
Grade 1 *	18	0.908 [0.876, 0.937]	0.616 [0.557, 0.674]	14.467 [8.892, 23.536]	0.871
Grade 2 *	17	0.599 [0.534, 0.661]	0.904 [0.872, 0.930]	13.971 [8.364, 23.335]	0.859
Grade 3 ^#^	10	0.786 [0.590, 0.917]	0.998 [0.987, 1.000]	150.220 [46.145, 489.000]	0.983
Tumor size, less than 2 cm					
Grade 1	4	0.852 [0.771, 0.913]	0.675 [0.509, 0.814]	15.319 [5.915, 39.677]	0.841
Grade 2	4	0.667 [0.498, 0.809]	0.844 [0.762, 0.906]	13.093 [5.143, 33.332]	0.834

CI, Confidence interval; OR, Odds ratio; AUC, Area under curve; SROC, summary receiver operating characteristic. *, reference: grade 3; ^#^, reference: grade 1 and 2.

## Data Availability

Not applicable.

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
