# Peer review of "Diagnostic Accuracy of ki-67 Labeling Index in Endoscopic Ultrasonography-Fine-Needle Aspiration Cytology and Biopsy of Pancreatic Neuroendocrine Neoplasms"

_diagnostics, 2023, doi:10.3390/diagnostics13172756_

Round 1

Reviewer 1 Report

The current paper is a meta-analytical review to evaluate the concordance rates of the Ki-67 labelling index between EUS-FNAC/FNB and surgical specimens. The authors also tried a subgroup analysis based on tumor grade, sampling method, and tumour size.

The present study is well-designed, and the analysis was good. The table, discussion and conclusions appears to be fine.

Author Response

Editor’s comment

Please note the need to extend the word count of the main text to at least 4000 words as you revise your manuscript.

Response) To the best of our knowledge, we have added to the text.

Reviewer 1.

The current paper is a meta-analytical review to evaluate the concordance rates of the Ki-67 labelling index between EUS-FNAC/FNB and surgical specimens. The authors also tried a subgroup analysis based on tumor grade, sampling method, and tumour size.

The present study is well-designed, and the analysis was good. The table, discussion and conclusions appears to be fine.

Response) Thank you for your careful review.

Reviewer 2 Report

Congrats for your study

Author Response

Editor’s comment

Please note the need to extend the word count of the main text to at least 4000 words as you revise your manuscript.

Response) To the best of our knowledge, we have added to the text.

Reviewer 2.

Congrats for your study

Response) Thank you for your careful review.

Reviewer 3 Report

1. A schematic diagram can be included for easier understanding of the findings and possible applications.

2. The conclusion section should be rewritten and include future directions.

Author Response

Editor’s comment

Please note the need to extend the word count of the main text to at least 4000 words as you revise your manuscript.

Response) To the best of our knowledge, we have added to the text.

Reviewer 3.

  1. A schematic diagram can be included for easier understanding of the findings and possible applications.

Response) Our study is a meta-analysis of the diagnostic accuracy of ki-67 labeling index in endoscopic ultrasonography-fine needle aspiration of pancreatic neuroendocrine neoplasms. As recommended by the reviewer, a schematic diagram is unfortunately difficult to add.

  1. The conclusion section should be rewritten and include future directions.

Response) We have added comments about future directions, as pointed out by the reviewer.

Round 2

Reviewer 3 Report

The revised manuscript looks better.